# Nutritional Issues in Head and Neck Cancer Patients

**DOI:** 10.3390/healthcare8020102

**Published:** 2020-04-17

**Authors:** Federico Bozzetti, Paolo Cotogni

**Affiliations:** 1Faculty of Medicine, University of Milan, 20122 Milan, Italy; federicobozzetti@gmail.com; 2Pain Management and Palliative Care, Department of Anesthesia, Intensive Care and Emergency, Molinette Hospital, University of Turin, 10126 Turin, Italy

**Keywords:** head and neck cancer, nutritional support, enteral nutrition, parenteral nutrition, oral nutritional supplements

## Abstract

The purpose of this paper is to update the oncologist on the correct approach to the nutritional care of the head and neck cancer patient. Recent scientific contributions on this issue, with a special emphasis on international guidelines and randomised clinical trials (RCTs), are reviewed. The following points are noteworthy: 1. Despite the advances in early diagnosis and modern treatment of head and neck cancer, this tumour still ranks first regarding frequency and severity of weight loss, both at the clinical presentation and during the therapy. 2. This is due to the combination of poor alimentation because of the tumour mass localization, as well as of the presence of an inflammatory response which furtherly drives catabolism. 3. Several studies have shown a very limited role for a dietary counselling unless it includes oral nutritional supplements which are protein or omega-3 fatty acid enriched. 4. A parental nutritional supplementation could represent an acceptable short-term alternative. 5. Long-term nutritional support relies on the use of percutaneous endoscopic gastrostomy (PEG), whereas the role of a prophylactic or “a la demande” PEG is still unsettled and requires further investigations. In conclusion, the nutritional approach using specific formulas and the appropriate route of administration should be part of the therapeutic armamentarium of the modern oncologist.

## 1. Introduction

Patients with head and neck cancer (HNC) are at increased risk of involuntary weight loss (WL) [1]. Cumulative weight losses may exceed 10% of pretreatment body weight [1,2,3] and WL of this magnitude is clinical as it contributes to prolonged hospitalization, forced treatment breaks [4], ineffective treatment response and impaired functional performance status, finally leading to reduced quality of life (QoL) and significantly lower survival [4,5,6,7,8,9,10,11]. In a large multivariable analysis of 1756 HNC patients [12] controlled for age, sex, cancer stage and site and performance status, both the weight loss grading system (a grading system based on combinations of body mass index and WL [13]) and dietary intake categories independently predicted overall survival.

Cancer-associated WL depends on a combination of inadequate dietary intake and metabolic abnormalities. In HNC patients, inadequate dietary intake is consistently related to nutrition impact symptoms [14,15]. Naïve HNC patients present during oncologic treatment a variety of symptoms (dysphagia, odynophagia, and anorexia) which lead to insufficient dietary intake and declining body weight [9]. A recent large study [12] reported that independent predictors of WL in naïve HNC patients included stage, performance status, and dietary intake.

During typical treatment of radiation therapy (RT), chemotherapy (CT) or both (CRT), mucositis, xerostomia, dysphagia, dysgeusia, and depression become exacerbated, making dietary intake and weight maintenance extremely challenging [16,17] and leading to the withdrawal of treatment in 20% of patients [18]. A large study of HNC stage I and II patients that were candidates of RT has shown that the following eight factors were significantly associated with a greater WL: all HNC sites other than the glottic larynx, TNM stage II disease, higher pre-RT body weight, dysphagia before RT, higher mucosa adverse effect of RT, lower dietary energy intake during RT, lower score of the digestive dimension on the Head and Neck Radiotherapy Questionnaire and a higher score of the constipation symptom on the European Organization for Research and Treatment of Cancer Quality of Life Questionnaire Core 30 (EORTC QLQ-C30) during RT [19]. More recently Mallick et al. [20] reported that in a multivariate analysis of 103 patients, the total planning target volume, prescription dose planning target volume and use of chemotherapy were significant correlated with a ≥5% weight loss after controlling for other factors. Patients could be risk-stratified based on the use of CRT and large planning target volumes: those with none, one or both factors had a likelihood of >5% weight loss of 0%, 30.3%, and 56.9% and likelihood of nasogastric tube placement of 5.3%, 15.2% and 37.3%, respectively.

After completing treatment, HNC patients may continue to be plagued by ongoing nutrition impact symptoms, which often result in inadequate dietary intake and weight loss [17,21] lasting for several months. Neuromuscular fibrosis, a recognised late effect of radiotherapy, may contribute to the development pharyngo-oesophageal strictures and chronic dysphagia. Its true prevalence is unknown because the manner in which dysphagia is assessed appears inconsistent, with patient weight and tube feeding dependence often serving as a surrogate in place of a formal swallowing assessment with a qualified speech pathologist. Using the MD Anderson dysphagia inventory as a measure of day-to-day swallowing function, patients who received nasogastric tube feeding (NTF) during treatment consistently scored higher across all domains (*p* < 0.001) [22]. Muscle atrophy associated with disuse is also implicated in the aetiology of chronic dysphagia, and seems proportional to the duration of tube dependence, late return to oral intake, and consequent long period of disuse.

Finally, in combination with the reduction of energy intake due to previously mentioned factors, the presence of cancer is associated with an alteration of metabolism caused by a systemic inflammation which adversely interferes with the utilisation of nutrients. Interestingly, some mediators involved in the metabolic derangements are also responsible for the onset of anorexia. Thus, HNC patients have, when compared with patients with other primary tumours, many reasons to become underfed: the involvement of structures devoted to food intake, tumour-dependent anorexia, and finally the local toxicity of the oncologic therapy in addition to the common metabolic derangements. In conclusion, cancer-associated WL is driven by reduced food intake and altered metabolism, and amongst these two factors, reduced dietary intake is likely the predominant factor [12,23].

## 2. Recognizing Potential Candidates for a Nutritional Intervention

The evaluation of the nutritional status of the patient should normally be part of the usual clinical examination, hence the oncologist should be able to easily identify the symptoms (anorexia, dysphagia, asthenia, poor performance status etc.) and signs (emaciated aspect, weight loss, depleted muscle and fat mass, etc.) which characterize the state of present or impending malnutrition. However, if a multidisciplinary team takes care of the HNC patient, criteria for planning a nutritional intervention should be rather homogeneous and objective. For this purpose, there is literature of several nutritional scores which quantifies the risk of morbidity due to a deteriorated nutritional status. None is better than the other, and the most used are those that are very simple and rapid such as the malnutrition universal screening tool [24] and malnutrition screening tool [25]. One of the most credited is the nutritional risk screening (NRS 2002) [26] which also relies on the clinical benefit (demonstrated in literature) for patients identified at nutritional risk receiving nutritional support. However, regardless of the nutritional status of the patient at the clinical presentation, the HNC oncologist is aware that treatment per se can affect the oral intake of food, and should be prepared to start with nutritional support before a critical deterioration of the nutritional state occurs.

## 3. How to Plan Nutritional Support

Type of nutritional support depends on three main factors:Is the inability to eat partial or complete?How long is the nutritional support expected to last?Is the oral route available and working?

There are at least three different clinical scenarios:

A. If the patient is able to eat by mouth, first, the oncologist should consider optimizing the usual diet. Quite often such dietary advice should be combined with prescription of oral nutritional supplements (ONS) because if left alone, the effectiveness is poor.

The choice of ONS should follow some general rules [27,28]:Choose ONS with a high protein content per mLChoose ONS enriched with leucine (or branched-chain amino acid) and omega-3 fatty acidsConsider that the compliance is better with high energy density (2 kcal/mL) ONSIncrease the variety of products according to the preference of the patient.As a rule, administer the ONS between meals and, alternatively, identify the best times and offer small amounts x multiple times (if bolus administration is not possible).

There are a variety of ONS products available. The choice will depend on patient preference, current macro- and micro-nutrient intake and local policy. Avoiding to propose a list of commercial products, we suggest, in addition to nutritional counselling, 2 bottles/day (250 mL) of a ready-to-use energy-dense (2 kcal/mL), high-protein, omega-3 fatty acid enriched oral formula providing 500 kilocalories, 23 g of proteins and 1.9 g of omega-3 fatty acids.

More important, the ONS should be part of the bundle prescribed by the oncologist (with pain killers, antiemetic agents or mouth washing solutions etc.) because patients should not consider that their consumption is somewhat optional or has to be completed only if a regular meal is neglected. It is equally noteworthy that patients should avoid the opposite situation in which ONS are considered a substitute for a meal, resulting in the avoidance of eating by mouth solely because they already consumed the ONS. It is generally reported that clinical benefits on the Eastern Cooperative Oncology Group scale (ECOG) occur when the intake is in the range of 250–600 kcal/day for ≥5 weeks. Three randomised clinical trials (RCTs) investigated the clinical benefit of ONS. Cereda et al. [29] reported better weight maintenance, increased protein-calorie intake, improved QoL and better anticancer treatment tolerance. Vasson et al. [30] investigated the effect of immunonutrition consisting of an arginine, omega-3 fatty acid, nucleotides-enriched diet on nutritional status and functional capacity in HNC or oesophageal cancer patients undergoing radiochemotherapy and found that functional capacity measured by ECOG scale and Karnofsky index was maintained in supplemented patients but significantly reduced in controls patients. Finally, Fietkau et al. [31] found that patients randomised to receive ONS containing omega-3 fatty acids had some benefit in subjective parameters as the Kondrup score and the subjective global assessment score.

B. If the patient is almost totally aphagic the nutritional support may be administered through parenteral nutrition (PN) if it is short-term (very few weeks) and the patient is hospitalized, and through tube feeding if nutritional support is required for longer periods or if the patient is at home. If the patient is at home, tube feeding is certainly more practical than intravenous nutrition, unless a “supplemental” intravenous nutrition is planned which can be delivered in a 6–8 h period to the patient, provided that they are partially able to receive oral nutrition in the outpatient department or at home.

There are two main questions regarding the nutritional intervention in this setting: first, whether a NTF or a percutaneous endoscopic gastrostomy (PEG) is better, and secondly, whether the early prophylactic feeding of patients at risk of further dysphagia or malnutrition is better than an intervention “a la demande”, the so-called reactive PEG (rPEG).

A few small RCTs showed that NTF [32] and early PEG [33,34] maintained weight or prevented weight loss better than optimal oral nutrition alone. Corry et al. [35] showed a benefit for weight in PEG versus NTF patients only at 6 weeks, but no difference at 6 months with overall QoL scores and complication rates being similar. Meta-analyses reported no significant differences in the overall complication rates between NTF and PEG [36], despite more frequent dislodgement in NTF patients and late dysphagia in PEG patients [37]. Discrepancies among the results of the RCT may depend on the heterogeneity of the nutritional regimens of the control groups as well as a lack of correct stratification of the patients by risk scores for malnutrition and hypophagia [19,20,38].

Regarding the effects of prophylactic PEG (pPEG) versus rPEG, a larger study [39] showed no difference between the two procedures in body weight, QoL and survival, and the result of a systematic review was also similar [40]. However, a more recent review [41] reported that while the pPEG strategy was associated with decreased malnutrition during treatment and improved QoL at six months, it was also associated with higher rates of long-term PEG dependence. The timing of PEG placement was not associated with improvement in tumour control or overall survival.

C. Many patients are in an intermediate condition—that is—they have an intermittent hypophagia which allows only a partial intake of food and supplements by mouth. For these patients, a practical solution is combining oral food/ONS with a supplemental PN which can be delivered daily or every other day. For patients suitable for a regimen of supplemental parenteral nutrition, it is important to know that there is a broad availability of nutritive admixtures on the market. There are 600-mL bags containing 600 nonprotein kcal and 35 g amino acid, which can cover approximately 40% of the macronutrient requirement of a medium-size patient and can be delivered in about eight hours in a central vein.

## 4. The Nutritional Regimen

The measured resting energy expenditure (REE) of HNC patients is around 22 kcal/Kg/day [42,43]. However, it is possible [44] that changes in the REE follow a U-shaped curve, with metabolic rates being highest at the beginning of treatment, end of treatment, and two weeks after treatment completion, and remaining high at one month after treatment completion compared to baseline [45]. De Souza et al. [46] studied 140 HNC patients and found that 57% of patients were hypermetabolic. The mean REE was 27 kcal/Kg/day (range 17–39) and the following equation was proposed to estimate the REE: 1042.34 + (124.28 × sex) − (10.08 × weight) + (19.32 × fat − free mass) where gender is 1 for male and 0 for female; weight is in Kg and fat-free mass is the value obtained by bioimpedance in Kg.

According to the experiences of some authors [12,14,47] an adequate regimen for a full nutritional support able to meet the requirements of the total energy expenditure should include ≥35 kcal/Kg/day and ≥1.5 g protein/Kg/day.

## 5. Recent Perspectives

Notably, there is a synergism between nutrition and physical exercise and the modern approach strongly suggests combining both interventions to nutritionally rehabilitate weight-losing patients [48]. A preliminary RCT combining nutritional support, anti-inflammatory agents and physical exercise [49] is under way. Also, there is evidence that most HNCs have a glycolytic phenotype and in these patients the use of a ketogenic diet to support the patient while reducing nutrients to the tumour appears attractive [50].

## 6. Conclusions

It is clear that negative effects of starvation and malnutrition are much better documented in literature than the potential benefits of nutritional support. Such evidence should not divert the attention of the oncologist to the nutritional issues of HNC patients for several reasons.First, it is easier to find significant correlations between malnutrition and poor outcomes in large retrospective series than to get significant results from small groups of patients enrolled within prospective RCTs comparing nutritional support versus controls which usually require a long time to be performed.For ethical reasons, in many studies, control groups always maintain some nutritional support and are not severely malnourished which considerably weakens the power of comparative studies.More importantly, the “treatment” arm in the published studies almost never received an optimized regimen. Current research emphasizes the need of a high-protein regimen with a special reference to protein-enriched, branched-chain amino acid and leucine [51]. Furthermore, it appears clear that even the optimal diet should be potentiated with the concurrent administration of anabolic and anticatabolic agents [48,51], but this approach has been little explored so far in HNC patients [52,53,54] despite promising preliminary results [30,31,53].

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
