# Peer review of "Nutritional Issues in Head and Neck Cancer Patients"

_healthcare, 2020, doi:10.3390/healthcare8020102_

Round 1

Reviewer 1 Report

Excellent article by a known expert in the field.

1. Perhaps a little more description of the types of nutritional supplements (a table) would add value to the manuscript. It would help the reader to apply this to practice more easily. For example: Section 3 is great in explaining the exact requirements, but often leaves the reader in a fix as to where they can find such a mix. More description would certainly help the reader. 

Author Response

Reviewer: 1

  1. Perhaps a little more description of the types of nutritional supplements (a table) would add value to the manuscript. It would help the reader to apply this to practice more easily. For example: Section 3 is great in explaining the exact requirements, but often leaves the reader in a fix as to where they can find such a mix. More description would certainly help the reader.

R. We thank the reviewer for this suggestion. Section 3 has been revised as suggested by adding some sentences (page 3, lines 115-119).

Reviewer 2 Report

In this review paper Authors report the recent scientific contributions on the correct approach to the nutritional care of the head-neck cancer patient, a type of tumour that still ranks first about the frequency and severity of weight loss.

The authors seem titled for the topics most covered in this review, and the paper appears quite informative about the main aspects of these problems, such as the poor alimentation because of the tumour mass features and inflammatory response. They also reviewed several studies about the role for dietary counselling with or without oral nutritional supplements with protein or omega-3, as well as parental nutritional supplementation.

They also underline a posible synergism between nutrition and physical exercise and the anti-infiammatory treatment.

Notably, studies about the very current use of ketogenic diet to support the patient while reducing nutrients to the tumour are also mentioned.

What does the abbreviation WLG indicate in row 41?

Author Response

What does the abbreviation WLG indicate in row 41?

R. Thank you. We eliminated the typo (page 2, line 41).
